# Development of an Objective Portable Measurement Device for Spinal Joint Accessory Motion Testing

**DOI:** 10.3390/s20010100

**Published:** 2019-12-23

**Authors:** Hsiao-Kuan Wu, Hung-Jen Lai, Ting Teng, Chung-Huang Yu

**Affiliations:** 1Department of Physical Therapy and Assistive Technology, National Yang-Ming University, Taipei 11221, Taiwan; kuan0728@gmail.com; 2TEH LIN Prosthetic & Orthopaedic Inc., New Taipei 24890, Taiwan; lai_ot@yahoo.com.tw; 3Department of Physical Medicine and Rehabilitation, National Taiwan University Hospital, Taipei 10002, Taiwan; ting15209@gmail.com; 4Preventive Medicine Research Center, National Yung-Ming University, Taipei 11221, Taiwan

**Keywords:** joint accessory motion test, manual therapy, spinal mobility, spinal stiffness

## Abstract

Joint accessory motion testing (JAMT) is a standard procedure used by manual therapists to assess and treat musculoskeletal disorders. Joint accessory motion (JAM) is movement that occurs between joint surfaces, and can be induced by applying force. The motion amount, end feel, symptoms, and resistance perceived by therapists during test procedures are recorded as evidence for the diagnosis, prognosis, treatment decision making, and intervention outcome. However, previous studies have shown that accessory motion tests have insufficient reliability. Recently, many instruments have been developed to increase test reliability, but these instruments quantify the test results with a single probe and utilize the external environment as a reference. Therefore, the measured displacement amount may be affected by other spinal segments. This study proposes an objective portable measurement device with two indenter probes for spinal JAMT, wherein the JAM was quantified by displacement and force measurements between two bones. The instrument was verified with a homemade spinal simulator and computer simulation. The results showed that the force-displacement curves measured by the JAMT device (JAMTD) and those simulated by the computer model exhibited similar characteristics. Moreover, a two-probe measurement could distinguish the differences in stiffness better than a one-probe measurement.

## 1. Introduction

The spine is made up of a series of bones that are stacked like blocks on top of one another with intervertebral discs in between. The spinal bones serve as a pillar that supports the weight of the body, and the joints link the spinal bones together and allow the torso to move. Thus, spinal joints are subjected to prolonged loading and repetitive high-speed impacts during daily dynamic activities. Overuse or diseases can cause joint injury and result in widespread pain and mobility loss. Pinpointing the injured joint is essential for proper treatment [1,2]. Joint accessory motion testing (JAMT) is one of the most common physical examinations used to assess the integrity and mobility of individual spinal joints.

Joint accessory motion (JAM) is the movement, such as spinning, rolling, and gliding, that occurs between joint surfaces [3]. JAM is required for full-range and pain-free physiological motion, but is not under direct voluntary control [4]. In the clinic, JAMT is often performed manually; i.e., evaluators exert forces on the linked bones and simultaneously discern the resistant force and displacement of the accessory motion [2]. The evaluators stop exerting force when reaching the end range of the accessory motion. The end range of the accessory motion is the feeling of a rapid increase in resistance with no further displacement occurring. After testing all spinal joints, the evaluator classifies the joint as normal, hypermobile, or hypomobile. The evaluator might draw a joint movement diagram to describe the abnormal joint [5,6].

Although experienced therapists could determine joint pathology through JAMT, reliability and reproducibility within and between evaluators are poor [7,8,9,10,11]. Therapists often overestimate the displacements and underestimate the forces applied [12].

To solve this problem, several devices have been developed to measure the displacement and loadings while performing JAMT [13]. In terms of displacement measurement, some devices feature a single probe, whereas others include multiple probes. Single-probe devices [14,15,16,17,18,19,20,21] measure displacements with respect to the external environment. Thus, several factors, such as the displacement of adjacent joints and the compression of layered soft tissues, are included in the displacement measurement, resulting in overestimations of the accessory movement [22,23,24,25]. On the other hand, properly designed multiple-probe devices can directly measure the displacement between the bones of a spinal joint. The design by Lee is a good example that measures displacement between the bones by subtracting the measurements of different probes.

The applied forces during JAMT can be easily recorded with force sensors. However, the manner in which the force is applied has an impact on the test results. Some devices employ electrical actuators to regulate the exerted forces [14,15,16,17,18,19,26]. This method has the advantage of keeping the applied forces consistent. However, the evaluator needs to learn how to set the maximum force value to prevent overloading the joint. Due to the requirement of a stable platform, these kinds of devices are often bulky, and are not convenient for clinical applications. The therapist needs to carefully adjust the device to measure different joints. Additionally, a bulky device might cause psychological stress to the subject and undue muscle contraction, which might lead to incorrect assessments [21,24,27]. More importantly, regarding safety concerns, the applied forces may not be sufficiently large to explore the full range of the joint; otherwise, the joint may be injured. On the other hand, when the force is applied manually [20,21], the evaluators can directly feel the response of the joint and extend the force up to the end range of a joint. Additionally, if the devices [27,28] were handheld, they would have the advantages of portability and flexibility, enabling quick changes in the testing location. However, as Kawchuk’s study showed, handheld devices are susceptible to inducing angulation errors and unsteady motion in displacement measurements.

In light of the design features discussed above and the need for a convenient, objective, and reliable clinical assessment device for JAMT, the aim of this study was to develop a manual handheld JAMT device (JAMTD) that is equipped with multiple probes to accurately measure displacements in joints while allowing evaluators to feel the joint response and exert the desired force. Moreover, this device should account for the errors caused by inclination and unsteady hands, thereby providing reliable and consistent results.

## 2. System Design

The JAMTD developed in this study contained three parts. The first part was the measurement front end, which contained a data acquisition (DAQ) system and sensors to measure and record the testing data. The second part was the real-time graphics display and data-relay module, which was an Android smartphone running a self-developed app. The third part was the data storage and detailed data analysis station, which comprised a personal computer (PC) running a proprietary LabVIEW program. The three parts were connected via Bluetooth and WiFi technology; the entire system structure is shown in Figure 1.

### 2.1. Multiprobe Measurement Front End

The measurement front end could be further explained in three aspects: the mechanical design, the sensing subsystem, and the integration platform.

#### 2.1.1. Mechanical Design

The mechanical structure of the device, as shown in Figure 2, included measuring and reference probes that were linked by a 55 mm-long linear slider. The measuring probe was attached to a gripping handle, and its position was adjustable from 15 to 85 mm along the handle. The reference probe was connected to a compartment accommodating the power source and the integration platform, and served as dead weight (5.20 N) for the reference probe. Two measuring pads were linked to the bottoms of the measuring and reference probes via ball-and-socket joints, which enabled the probes to fit on the contact points for a contoured surface, and to accommodate device tilting.

#### 2.1.2. Sensing Subsystem

The sensing subsystem included several sensors for various purposes. A miniature S-type load cell was equipped in the measuring probe and was used to measure the force applied by an evaluator. Previous studies have indicated that the safe range of applied force for asymptomatic subjects was 135 N or 200 N [14,25,29], and 105 N for symptomatic subjects [14]. Therefore, a load cell with a 200 N measuring range was selected (Tecsis Sensors Co., Ltd., Shenzhen, China). A tactile switch set in the middle of the reference probe was used to detect whether the probe was in contact with the subject. An optical linear encoder from US Digital (US Digital, Washington, DC, USA) with a linear strip resolution of 500 counts per inch was used to measure the displacement between the probes. The length was sufficient to accommodate the maximum displacement in the lumbar joint, i.e., 13 mm, and the level difference between spinal bones. Another miniature S-type load cell was utilized as the hand grip gauge for the user to indicate his/her pain level during testing [30]. A three-axis accelerometer was used as an inclinometer to measure the tilt angles; the function of this device is explained further in the following section.

#### 2.1.3. Integration platform

Figure 3 shows that a PSoC 3 (Cypress Semiconductor Corp., California, CA, USA) microcontroller was used to acquire data from the sensors mentioned above at 200 samples per second. The status of the measurement front end and the numerical data of the measurement results could also be immediately shown on a 16 × 2 character liquid-crystal display (LCD) module. A Bluetooth module was connected and utilized to transmit the acquired data to the smartphone (or tablet) for a graphical display. The firmware was able to transmit the data automatically when the reference probe was in contact with the subject. A mechanical button was used to trigger a calibration procedure, which is discussed in a later section.

### 2.2. Real-Time Graphics Display and Data-Relay Module

To provide instant feedback to the evaluator, a graphics display and data-relay module were implemented by a smartphone/tablet running a self-developed application program. Furthermore, to increase the operational consistency, an auditory cue of 0.5 Hz tempo was provided by the module while performing the JAMT. The JAMs were presented in the format of a “force-displacement graph” for each joint in real time, as shown in Figure 1. Moreover, a warning sound could be issued when the exerted force reached the safety limit, or the pain level of the patient reached an unacceptable range. Above all, the module also served as a data-relay module that retained the newly-acquired data and uploaded the information to the file server, which is discussed in the following section.

### 2.3. Data Storage and Detailed Data Analysis Station

The data temporarily retained in the data-relay module were sent to a PC for permanent storage and further analysis. A self-developed LabVIEW program (National Instruments Corporation, Austin, TX, USA) allowed the user to retrieve time-series data, postprocess the raw data, and analyze the characteristics of the force-displacement curves.

## 3. Correction of Displacements and Forces from Unsteady Hand Movement

### 3.1. Nullifying the Additional Readings Caused by Unsteady Hand Movement

Additional readings caused by the angulation of an unsteady hand are shown in Figure 4. Figure 4b shows the contact points on the same hard level surface; when the probes were perpendicular to the surface, the encoder reading $R_{i}$ was the same as the encoder offset $D_{0}$. However, when the probes were tilted along the line of the contact points (i.e., around the axis), there was an additional reading $D_{e0}$, as shown in Figure 4c. Moreover, when the contact points were not on the same level and the probe was tilted, as shown in Figure 4d, e (i.e., while performing JAMT), the measuring probe was further extruded to make its vertical component equal to $D_{PA}$. As a result, the actual posterior–anterior (P–A) displacement could be calculated from Equation (3). The P–A force can be calculated from the axial force with the direction cosine projection using Equation (4), as shown in Figure 5.
(1)$$D_{e0}=G\tan\theta_{i}$$
(2)$$D_{i}=\;R_{i}-D_{0}-D_{e0}=R_{i}-D_{0}-G\tan\theta_{i}$$
(3)$$D_{PA}=D_{i}\cos\theta_{i}=\left(R_{i}-D_{0}-G\tan\theta_{i}\right)\cos\theta_{i}$$
where
G is the gap distance between probes (system parameter),$D_{0}$ is the encoder offset (system parameter),$\theta_{i}$ is the tilt angle, (a.k.a. the pitch angle in the 3D case, as shown in Figure 5$D_{PA}$ is the P–A displacement,$R_{i}$ is the encoder reading,$D_{e0}$ is the additional reading caused by tilt when the probes are on the same level, and$D_{i}$ is the extruded displacement along the tilt angle to the indented level.
(4)$$F_{PA}=F\cos\theta_{z}\;=\;(F_{m}-F_{0})\;\cos\theta_{z}$$
where
F is the applied force,$F_{m}$ is the measured force,$F_{PA}$ is the P–A component of the applied force,$F_{0}$ is the load cell offset, and$\cos\theta_{z}$ is the direction cosine along the Z axis.

### 3.2. System Parameter Identification Procedure

In clinics, the probe indentation and the gap between two probes might be changed to fit the patient’s spinal curve. To identify the system parameters G and $D_{0}$ without an additional ruler, a specific identification procedure was used. The identification procedure was as follows. First, the evaluator lifted the device in the air to obtain the offsets of the load cell and the origin of the linear encoder, as shown in Figure 4a. Then, the evaluator placed the device on a hard level surface, such as a bench table, and aligned the bubble level indicator on the device to ensure that the probes were vertical in order to obtain the misalignment of the accelerometer and the level offset $D_{0}$ from the encoder reading, as shown in Figure 4b. Finally, the device was tilted in all directions to a couple of tens of degrees as the DAQ system sampled the serial angles $\theta_{i}$ and related encoder reading $R_{i}$ to obtain the gap distance G using Equation (5) and least-squares linear regression, as shown in Figure 4c.
(5)$$G=\left(R_{i}-D_{0}\right)\cot\theta_{i}$$
where
G is the gap distance between probes (system parameter),$D_{0}$ is the encoder offset (system parameter), and$\theta_{i}$ is the tilt angle.

When the system started or the identification button on the JAMTD was pressed, the identification procedure of the system was executed to obtain the system parameters, i.e., the load cell offset, the accelerometer axis misalignment, and the distance between the probes.

### 3.3. Data Processing

The data processing procedure and results are shown in Figure 6. The acquired data were first smoothed by a Savitzky-Golay filter with a second-order polynomial to the data frame of 41 data points. The purpose of using the Savitzky-Golay filter was to increase the signal-to-noise ratio without substantially distorting the signal [31]. Next, the P–A component of force and displacement were calculated based on the system parameters.

## 4. System Verification

The system verification of the JAMTD included a single-spring test, a spinal simulation test, and an Adams computer modeling simulation. The single-spring test was used to verify the accuracy of the sensing system by measuring the spring stiffness with the JAMTD. The spinal simulator test was used to verify the ability of the JAMTD to distinguish differences in stiffness between two segments via one-probe and two-probe methods.

### 4.1. Testing Rig

We designed a test rig that was connected with vertical and horizontal springs to simulate the spine structure, as shown in Figure 7a. This homemade testing rig of spinal simulation, which was composed of five sliders fixed on an acrylic frame, was used to verify the JAMTD in this study. The sliders could be set to limit the maximum downward displacement by inserting stop screws or posts underneath. Different numbers and types of springs could be attached between adjacent sliders and between a slider and the frame; thus, the stiffness could be changed.

Furthermore, the vertical springs represent the cross-sectional overlying tissues, such as skin, muscle, fascia, and fat, that are affected by compression force. The horizontal spring represents the longitudinal bonding tissues around the joint, such as the ligaments, tendons, and muscles, which are tensioned to prevent joint dislocation and provide joint stability.

For ease of description, the arrangements of springs were represented as structure formulas, wherein the horizontal and vertical bars represent horizontal and vertical springs, respectively, and R and M represent the sliders where the reference probe and measuring probe could be applied, respectively. Figure 7 shows one kind of arrangement that was represented as –R |=–M| . In this arrangement, the reference slider (R) was linked to the frame by one horizontal spring and one vertical spring, the measurement slider (M) was linked to the frame by one horizontal spring and one vertical spring, and there were two horizontal springs linking the reference and measurement slides.

To verify the testing results from the testing rig, these simulation conditions were also modeled with Adams Student Edition (MSC Software Corporation, Newport Beach, CA, USA) to calculate the expected force-displacement behavior, as shown in Figure 7b. The simulation conditions were as follows: the applied force was quasistatic; the k value of each spring was 3.71 N/mm; without preload, the friction between each object was ignored; and the allowable displacement of the sliders was ten millimeters before hitting a hard surface.

### 4.2. Single-Spring Test

The developed JAMTD was first verified by comparing the measured stiffness with the given values of four arbitrary selected extension springs. Those four springs (I-IV) were produced by the Associated Spring Raymond Company, and the stiffness values of those springs were individually measured by the company with a commercial spring tester (Japan Instrumentation System Co., Ltd., Nara, Japan). To measure the spring stiffness with our JAMTD, the spring was attached to a measurement slider, and the frame of the spinal simulator with the $._{M}^{\;|}.$ spring arrangement and the reference slider was fixed. The statistical results of the stiffness coefficient K, which corresponds to the linear part of the force-displacement curve measured by the JAMTD, were based on seven measurements and are shown as the mean and standard deviation. A one-sample t-test was conducted on the stiffness coefficient K to evaluate whether those mean results were significantly different from the reference value. The alpha level was set to 0.05.

### 4.3. Spinal Simulator Test

To understand the applicability of our JAMTD to a real spine, the device was tested on the testing rig of different arrangements of springs with known parameters. The basic arrangement of springs was –R |−–M| , where one spring is between the probes and the frame. The spring between the probes was removed or added for another arrangement (i.e., the spring in the middle position: –R |..−M|  or –R |=–M| ). In addition, the maximal displacement of the slider was set to 10 mm by adjusting a stop screw on the slider. Two measuring methods, measuring with one probe or two probes, were applied in this simulation to distinguish the three different conditions. Each condition was tested five times, and is shown as the mean value.

The purpose of this spinal simulator test is to determine whether the two-probe measuring method is necessary to distinguish the differences in stiffness between adjacent sliders, which is an interesting part of JAMT.

## 5. Results of System Verification

### 5.1. Single-Spring Test

The results of the single-spring test are shown in Table 1. The stiffness values of the four springs (I-IV) measured by JAMTD were 3.109 ± 0.098, 4.305 ± 0.167, 11.336 ± 0.314, and 17.505 ± 0.314. These results were not significantly different from the reference values.

### 5.2. One-Probe Test and Simulation

The force-displacement curves obtained by one probe simulated by the Adams model and those measured by the JAMTD are shown in Figure 8a,b, respectively. These two force-displacement curves were highly similar and exhibited the following characteristics. First, due to the constant force applied to the reference slider, the displacement started from a negative value. As the applied force increased, the displacement increased until the measuring probe reached the limit.

The test with an additional spring attached between the two sliders (−R |=−M| ) exhibited progressively steeper curves (larger stiffness) than the test with the basic arrangement. By contrast, the test with no spring attached between the two sliders (−R |..−M| ) exhibited a gentler curve (smaller stiffness) than the test with the basic arrangement.

### 5.3. Two-Probe Test and Simulation

The force-displacement curves obtained by two probes simulated by the Adams model and those measured by the JAMTD are shown in Figure 9a, b, respectively. In particular, the displacement difference between the sliders in the three arrangements could be observed only with the two-probe measurement method. The test with an additional spring attached between the two sliders (−R |=−M| ) had a shorter displacement than the test with the basic arrangement, and the test with no spring attached between the two sliders (−R |..−M| ) had a longer displacement than the test with the basic arrangement.

The data acquired by the LabVIEW program are shown in Figure 10. The force-displacement curves filtered by a Savitzky-Golay filter are shown in Figure 10a. The P–A component of the filtered data is shown in Figure 10b. The separated variables in the time-series data, including force, displacement and roll, and pitch and tilt angle ($\theta_{z}$), are shown in Figure 10c. The definitions of the tilting angles are shown in Figure 5.

## 6. Discussion

### 6.1. Instrument Design

In previous studies, the evaluator might need to learn new methods in order to use motor-driven devices, such as setting the maximum force value to prevent overdose and adjusting the platform in order to place the probe [15,16,26,29,32]. Our manual handheld JAMTD allows the evaluator to directly place the probes on the spinal joints after palpation, as with traditional methods. The evaluator could also feel the resistance and displacement of the joint accessory movement when applying the force to the joint. In addition, our JAMTD provides a quantitative visual display of the force and displacement in real time, and the evaluator could visualize his/her assessment to avoid underestimating the applied force. The handheld design of the JAMTD allowed the evaluator to feel the patient’s response, and subjectively recorded the exerted force and displacement. Furthermore, regarding safety, the JAMTD issues a warning sound when the exerted force reaches the preset safety limit. The device also provides a hand grip gauge for the subject to hold to express his/her level of pain. When the subject feels pain, he/she squeezes it. The more pain the subject feels, the harder he/she squeezes. When the pain scale reaches the preset limit, the device warns the evaluator. Thus, this device is convenient and safe for clinical applications.

The JAMTD could objectively measure the resistance and displacement of the joint, and automatically record the spinal stiffness. The evaluator does not need to record the subjective feeling of joint mobility and add that information to the clinical chart. Moreover, the device equipped with two probes could measure the displacement in the joint and minimize the influence of environmental factors, such as the hardness of the treatment bed.

Our JAMTD could account for errors caused by unsteady hands to provide reliable and consistent results. Changes in the tilt angle during a test did affect the displacement results, as a previous study suggested [33]. In this study, the additional displacement due to tilting was nullified by extracting the P–A displacement between the probes together with the P–A component of the applied force. The force-displacement diagram showed high uniformity.

### 6.2. Single-Spring Test

The system was verified by measuring the stiffness of four arbitrarily selected springs with our JAMTD. The verification results showed that there was no significant difference between the measured value and the reference value. Moreover, the results had a coefficient of variation that was less than 4%. The statistical results showed low variability.

### 6.3. Spinal Simulation Test

Spinal models in which the stiffness can be adjusted by changing springs have been used in previous studies to simulate and measure the displacement and applied force under different stiffness conditions of a spine [12,34]. Spinal models are used to test the reliability of manual JAMT devices or other spinal measuring devices. According to the position of the springs, the stiffness of the measurement should be constant during the testing process in previous studies. However, the actual stiffness of the joints gradually increases as the force increases [26]. A possible reason for this phenomenon is that the actual alignment of the ligaments or muscles restricted the end range of accessory motion of the joint; i.e., the alignment is perpendicular to the joint surface and the direction of the applied force. To simulate a similar condition for the spine, the horizontal spring of our spinal simulator was used to simulate the ligaments or muscles, whereas the vertical springs were used to simulate the P–A components of the body, such as the transverse of the muscle, fat, or skin, or the therapeutic bed.

Three spring arrangements on our spine simulator were chosen to simulate the possible conditions of the spinal joints, such as the upper or lower level of the spine tightness. The stiffness between adjacent bones was an interesting part during JAMT. Hence, the main issue of those simulation tests was distinguishing the −R |=−M| arrangement from other arrangements.

The tests with an additional spring attached horizontally had progressively steeper curves than the test with the basic arrangement because the vertical stiffness component of the horizontal spring gradually increased as the angle of the spring increased during the test. Hence, the influence of the horizontal spring was more prominent when the displacement increased. Therefore, the stiffness near the later range was more representative than the stiffness near the initial range in the accessory motion test. Moreover, the −R|=−M| test had the shortest displacement to the end range because the reference side was pulled down more by the additional spring attached at the middle of the two sides.

Although the force-displacement curves produced by the JAMTD and Adams simulation exhibited similar characteristics, different displacement offsets and bouncing curves at the beginning region were observed. These differences could be due to the preload of the springs, the friction between each object that was ignored, and the quasistatic motion condition in the Adams simulation.

The tilting movements of the hand were also observed from the angle change in the time-series data in Figure 10. The hand movement indeed caused the additional displacement reading. Moreover, the P–A displacement could be calculated, and the difference in stop limits was also greatly reduced. This can be verified by the results shown in Figure 10b, where the maximal displacement of the slider in P–A displacement under the no-spring condition was close to 10 mm, which was set by adjusting a stop screw on the slider.

## 7. Summary & Conclusions

An objective portable measurement device with two indenter probes for spinal JAMT was developed in this study. The JAM could be quantified by measuring the displacement and force between two bones. The instrument was verified by a homemade spinal simulator and computer simulation. The results showed that the force-displacement curves measured by the JAMTD and those simulated by the computer model exhibited similar characteristics. Moreover, the two-probe measurement method could distinguish the differences in stiffness better than the one-probe measurement method.

## Figures and Tables

**Figure 1 sensors-20-00100-f001:**
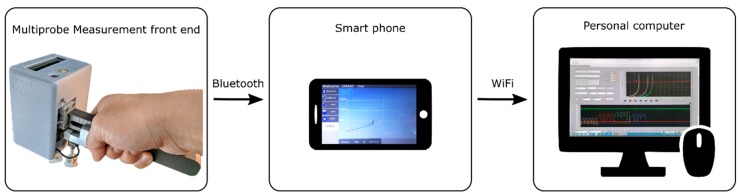
System structure.

**Figure 2 sensors-20-00100-f002:**
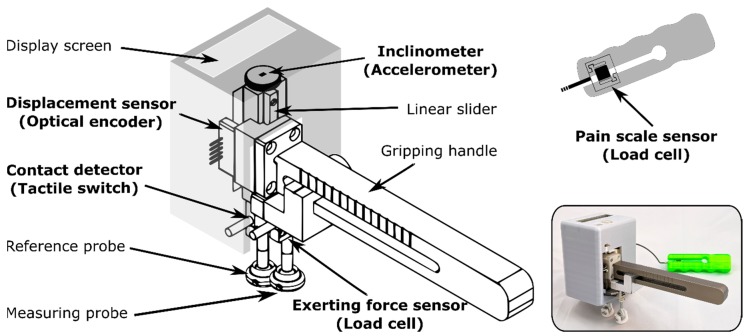
Mechanical structure of the measurement front end.

**Figure 3 sensors-20-00100-f003:**
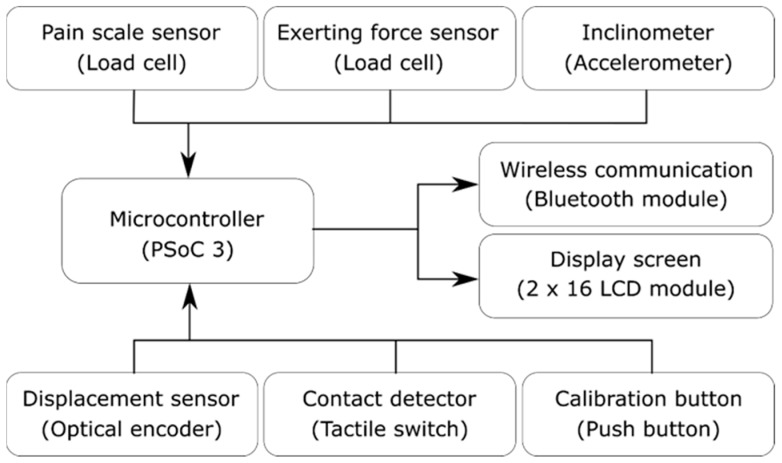
Sensing subsystem and integration platform of the measurement front end.

**Figure 4 sensors-20-00100-f004:**
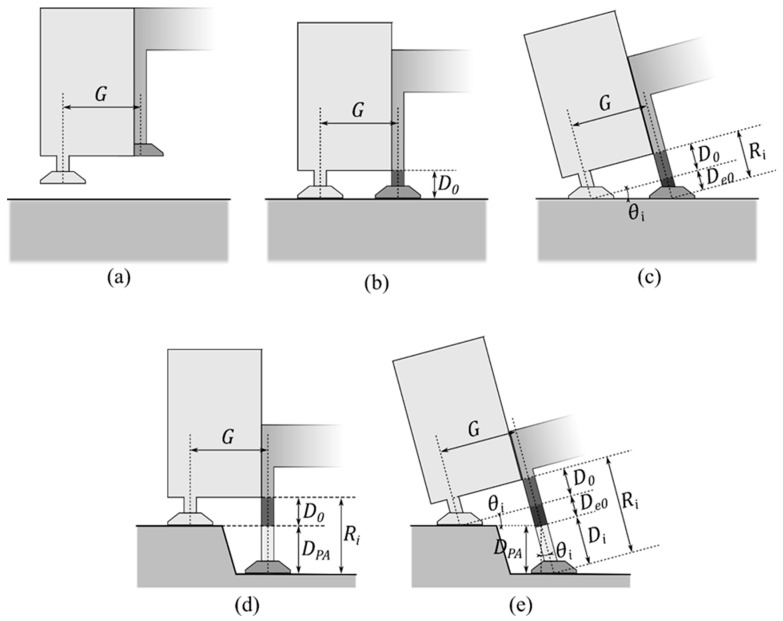
P–A component of displacement. (**a**) Device is lifted off the surface. (**b**) Device is laid on the hard level surface. (**c**) False reading caused by angulation of an unsteady hand when the probes are on the same level. (**d**) Contact points are not on the same level while exerting force. (**e**) False reading caused by the angulation of an unsteady hand when the probes are not on the same level.

**Figure 5 sensors-20-00100-f005:**
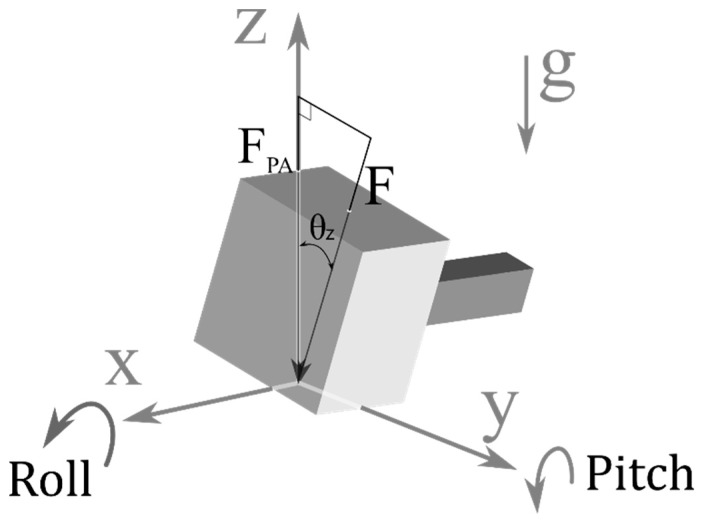
P–A component of force

**Figure 6 sensors-20-00100-f006:**
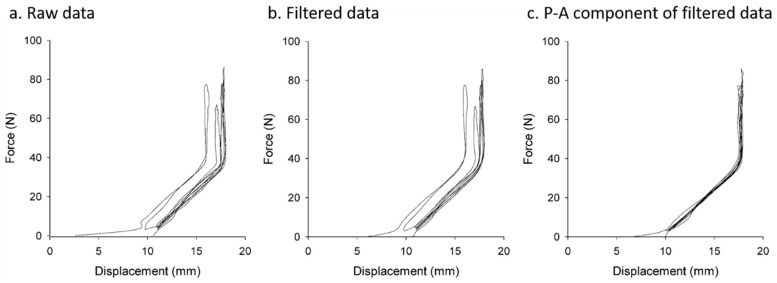
JAMT results through different data processing stages. (**a**) Raw JAMT data. (**b**) Filtered data smoothed by a Savitzky-Golay filter. (**c**) P–A components of force and displacement calculated based on the system parameters.

**Figure 7 sensors-20-00100-f007:**
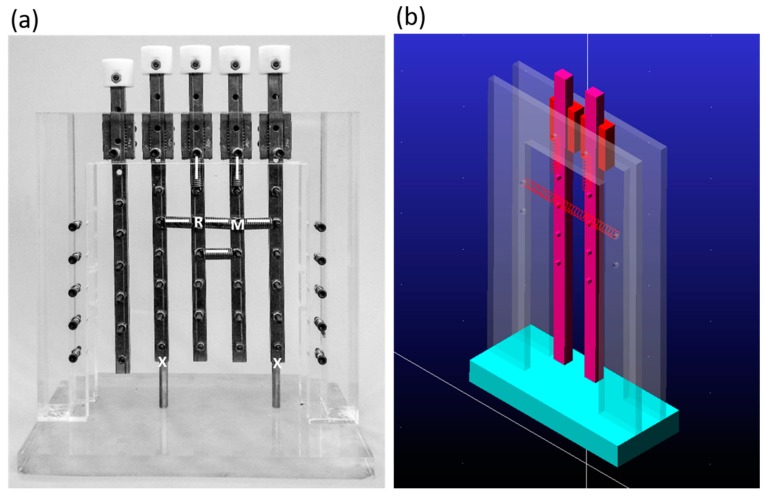
Spinal simulator (**a**) Spinal simulator with the –R |=–M|  arrangement of springs. (**b**) Simulated model built with Adams software.

**Figure 8 sensors-20-00100-f008:**
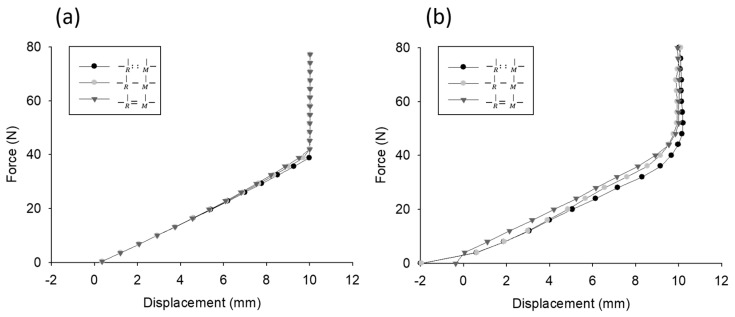
Simulation results of force-displacement curves measured by one probe. (**a**) Simulated by the Adams model. (**b**) Measured by the JAMTD.

**Figure 9 sensors-20-00100-f009:**
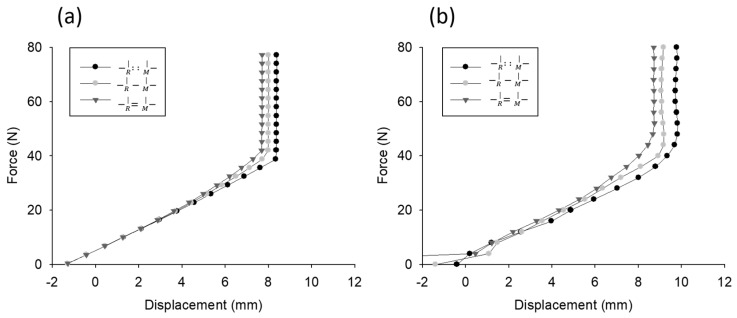
Simulation results of force-displacement curves measured with two probes. (**a**) Simulated by the Adams model. (**b**) Measured by the JAMTD.

**Figure 10 sensors-20-00100-f010:**
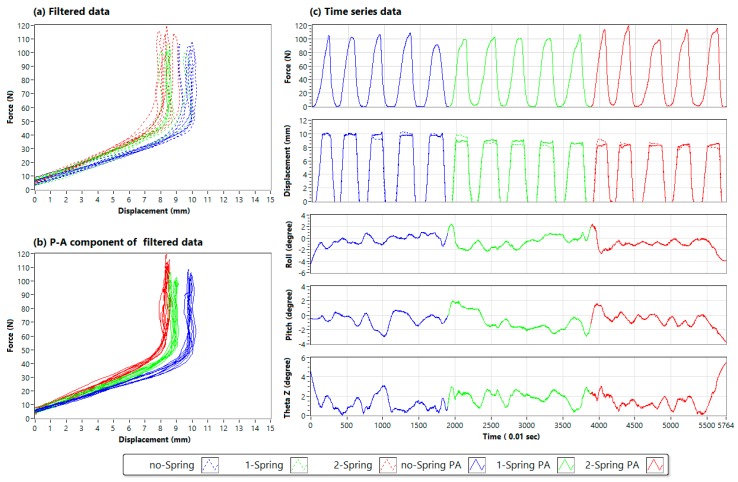
Detailed data measured in the two-probe test. (**a**) Filtered force-displacement curves. (**b**) P–A component of the force-displacement curves. (**c**) Time-series data (no Spring: filtered data from the test with no spring attached between the two sliders; 1 Spring: filtered data from the test with one spring attached between the two sliders; 2 Springs: filtered data from the test with two springs attached between the two sliders; no Spring PA: P–A component of the filtered data from the test with no spring attached between the two sliders; 1 Spring PA: P–A component of the filtered data from the test with one spring attached between the two sliders; 2 Springs PA: P–A component of the filtered data from the test with two springs attached between the two sliders).

**Table 1 sensors-20-00100-t001:** System verification results.

Spring	Reference Value ^1.3^	JAMTD Mean (SD) ^1^	95% CI	CV ^2^
I	3.177	3.109 (0.098)	3.011−3.197	3.2%
II	4.197	4.305 (0.167)	4.158−4.472	3.9%
III	11.376	11.336 (0.314)	11.042−11.621	2.8%
IV	17.553	17.505 (0.314)	17.211−17.789	1.8%

^1^ Values are shown in N/mm (reference, mean, SD, 95% confidence interval). ^2^ CV: coefficient of variation is given in % = (σ/mean stiffness) × 100. ^3^ The reference values were provided by the spring manufacturing company and were individually measured with a commercial machine.

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
