# Peer review of "Development of an Objective Portable Measurement Device for Spinal Joint Accessory Motion Testing"

_sensors, 2019, doi:10.3390/s20010100_

Round 1

Reviewer 1 Report

However, I have some serious doubts about the paper, which I will summarize in a few lines, following the structure of the article:

First of all, the paper lacks an up to date description of the state of the art that could give the reader an idea of the current systems used for measuring JAM. What is more, only three references are less than 10 years old (and they are from 2011). A general idea of recent designs presented in the literature is needed to evaluate the real contribution of this new device. Second, the Methods section intends to provide a bulk of information in a few pages, so some parts of it are not clear enough. For example: the calibration procedure establishes that the user needs to move the probe “a couple of tens degrees” in all directions. Is that angle measured somehow? If it is not, is it accurate enough for calibration? Also in the Methods Section, some basic functioning characteristics of the device need to be validated. As a first idea, the force applied by the system should be measured. So that the user is sure that the intended force agrees with the applied force. Then, the limit force to be applied should be tested. Given the challenge of the positioning of the device (in terms of angles of application and different levels in the surface), some sort of validation of those two variables should be presented rather than a force-displacement curve. For example, different users could apply the probe and variability in the measurements (probably due to different angles of application) could be reported. According to the issues mentioned before, the results presented do not show enough evidence that the system is accurate or reliable nor even safe for the intended purpose.

Reviewer 2 Report

This paper proposes an objective portable measurement device with two indenter probes for spinal joint accessory motion testing.

The paper is well structured, and generally well written.

In section 3 and in the following ones, there is an error in the numbering of the figures and therefore also in their citation in the test. Please check.

System verification was done with simulated data. If possible, it would be interesting to do the same verification with real data.

Round 2

Reviewer 1 Report

I appreciate the repply of the authors and the additions that they have provided. I still find a few issues with this article and it probably is because we have not understand each other:

1) It would be really important if the authors can explain why this work is important in the current state of the art. I understand that the authors found only three papers related to systems to measure JAM in the last decade, which may be understood as if there is no much interest in the topic nowadays. So, I would suggest that the authors include applications of similar systems in recent years, and I would not expect them to be only three in the past decade.

2) As I mentioned before, the force applied by the system should be measured. Given that your system not only measures displacement but also exerts a force to the vertebrae, and given this is a medical system, the exerted force need not only to be calibrated against standard weights, but it should also be actually measured in real applications.

This goes beyond the selection of the user, who may choose a "higher" or "lower" range. It is the system, as a secure medical device, which needs to apply a force accurately read and also, warn the user about forces that are reaching dangerous limits.

I am really sorry to say, but the rules about medical devices do not agree with this statement "the force is exerted by professional evaluators who can still feel the responses to the assessment and should be able to judge whether the force is beyond the safety limit." A system should not provide forces that are beyond the safety limit.

Round 3

Reviewer 1 Report

Dear authors,

Congratulations on the work performed on the article.

I would like to share with you my opinion regarding a medical system, in terms of safety, just as an exchange of viewpoints. It may be helpful for you to consider.

I understand that normally, the JAMT is a procedure performed entirely by the clinician and guided by his or her experience. However, he/she is the only responsible for exerting an adequate force, if they are the only force effectors. Once there is a piece of equipment to perform the force, even if it is only transmitted by the equipment, then there is a shared responsibility. And IF the clinician fails to evaluate the maximum force, then the system should be able to do it.

The evaluation of pain is an aftermath result and should rather be prevented, instead of use it as a warning sing.

In this respect, I still consider that your equipment requires an extra level of development that could be performed in future designs.